# Synthesis, Molecular Docking Analysis, and Carbonic Anhydrase Inhibitory Evaluations of Benzenesulfonamide Derivatives Containing Thiazolidinone

**DOI:** 10.3390/molecules24132418

**Published:** 2019-06-30

**Authors:** Zuo-Peng Zhang, Ze-Fa Yin, Jia-Yue Li, Zhi-Peng Wang, Qian-Jie Wu, Jian Wang, Yang Liu, Mao-Sheng Cheng

**Affiliations:** Key Laboratory of Structure-Based Drug Design & Discovery of Ministry of Education, School of Pharmaceutical Engineering, Shenyang Pharmaceutical University, Shenyang 110016, China

**Keywords:** human carbonic anhydrase IX, thiazolidinone, benzenesulfonamide, docking study

## Abstract

To find novel human carbonic anhydrase (*h*CA) inhibitors, we synthesized thirteen compounds by combining thiazolidinone with benzenesulfonamide. The result of the X-ray single-crystal diffraction experiment confirmed the configuration of this class of compounds. The enzyme inhibition assays against *h*CA II and IX showed desirable potency profiles, as effective as the positive controls. The docking studies revealed that compounds (**2**) and (**7**) efficiently bound in the active site cavity of *h*CA IX by forming sufficient interactions with active site residues. The fragment of thiazolidinone played an important role in the binding of the molecules to the active site.

## 1. Introduction

Human carbonic anhydrases (*h*CAs, EC 4.2.1.1) are important metalloenzymes that can catalyze the interconversion of CO_2_ and H_2_O with HCO_3_^−^ and H^+^. These enzymes play a pivotal role in many cellular and physiological processes, such as electrolyte secretion, pH homeostasis, respiration, and biosynthetic pathways. There are at least 15 CA isoforms in humans, which differ in molecular properties, subcellular localization, and tissue distribution [1,2,3,4]. Uniquely, CA IX has low expression in many normal tissues, but it is overexpressed in numerous solid tumors, assisting in tumor cell survival, proliferation, and metastasis in the tumor microenvironment [5,6,7,8,9,10,11]. Therefore, CA IX has been validated as a promising target for cancer diagnostics and treatments. Currently, there are a large number of CA IX inhibitors reported in the literature, but only SLC-0111 (Figure 1) has completed Phase I clinical trials [12,13,14].

In the area of medicinal chemistry, five-membered heterocyclic compounds, especially heterocycles having more than one hetero atom, have shown a diverse range of biological activities. Among them, thiazolidinone, which bears a sulfur atom at 1-position, a nitrogen atom at 3-position and a carbonyl group at 2-, 4-, or 5-positions, has received a great deal of attention over the years since it is a significant and versatile moiety that has occupied a prominent position in medicinal chemistry [15,16,17,18,19,20,21,22,23,24]. Derivatives containing a thiazolidinone moiety have demonstrated a wide range of pharmacological properties, including anti-HIV, anti-malarial, anti-inflammatory and hypoglycaemic activities [25,26,27]. In many cases, the introduction of thiazolidinone into molecules can enhance the interaction between the whole molecule and the biological target. In the field of anti-tumor drugs, thiazolidinone fragments are also frequently chosen to modify lead compounds. A variety of thiazolidinone-based derivatives with different substituents are being considered as potential anticancer agents [24,28]. In particular, previous work by the Azam group pointed out that introducing a five-membered heterocyclic ring in some compounds can help increase *h*CA IX inhibitory activity, achieving *K*i values in the low micromolar range [28]. Supuran and Akdemir’s work also drew the same conclusion [29].

To extend the applicability of thiazolidinone to CA inhibitors, in this work, we synthesized thirteen novel compounds by combining sulfonamide with thiazolidinone. These compounds were then tested for enzyme inhibition of *h*CA IX and off-target *h*CA II isoform (as a subtype of the CA family, CA II is widely distributed in the body and participates in various physiological activities). The inhibitory activities were determined to be in the low nanomolar region. To determine the binding mechanism and rationalize the trend of the inhibition profiles, molecular docking studies of the designed hybrids in the active site of *h*CA IX and II were carried out, respectively.

## 2. Results and Discussion

### 2.1. Chemistry

The general synthetic route for the synthesis of the target benzenesulfonamide-thiazolidinone derivatives is shown in Scheme 1. Isothiocyanatobenzene reacted with mafenide hydrochloride to afford benzenesulfonamide (**1**) as the intermediate, which continued in the next cyclization reaction to form compound (**2**). Thereafter, compounds (**3**–**14**) were obtained via the Knoevenagel condensation of compound (**2**) reacting with different substituted benzaldehydes.

### 2.2. hCA Enzyme Inhibition Studies

The synthesized compounds (**2**–**14**) were evaluated for inhibitory activity against *h*CA II and IX and by monitoring the hydrolysis of 4-nitrophenyl acetate (4-NPA). The inhibition data are summarized in Table 1.

According to the data in Table 1, the structure-activity relationships (SAR) of compounds (**2**–**14**) are preliminarily summarized below.

i. Compared to the lead compound mafenide hydrochloride, most synthesized compounds showed a significant increase in the inhibitory activity against both *h*CA II and IX, which were comparable to the positive control drugs AZM and SLC-0111, except for the compound (**12**) and (**13**). There was a certain relationship between the molecular docking binding scores with the value of IC_50_, and the compound with better in vitro activity owned more negative score value.

ii. Compounds (**3**–**14**) were synthesized by introducing substituted aromatic groups into the adjacent carbonyl group of thiazolidinone. In most cases, the introduction of an electron-withdrawing group into the benzenesulfonamide-thiazolidinone scaffold caused a clear decrease in the inhibitory potencies against *h*CA. Electron-donating groups provided better inhibition results, but when there was more than one electron-donating group, the activity of the compound decreased slightly as in compound (**14**).

iii. Among the halogenated benzene substituted derivatives (**4**–**6**) with the halogen at the meta position, the fluorine substituted derivative showed the highest inhibitory activity. Comparing compounds (**4**) and (**8**)–(**10**), we found that the compounds with substituents in the meta position showed better inhibition activity than the ones with substituents in the para position.

iv. The weakest compound among the series was the benzyloxy substituted compound (**12**). The isopropyl substituted compound (**13**) also exhibited poor inhibitory activity. Therefore, we can conclude that increased steric hindrance of the substituents decreases the inhibition of *h*CA. However, the effect of larger steric hindrance on the inhibitory activity of *h*CA II was not as great as that of *h*CA IX.

### 2.3. Docking Studies into the Active Site of hCAs

To further annotate the binding mode of the series compounds, compounds (**4**), (**7**), (**11**) and 5FL4-ref were chosen for docking studies into the active site of *h*CA IX. The results were shown in Figure 2.

As shown in part A of Figure 2, compound 5FL4-ref interacted with the active site of CA IX by one hydrogen bond between the Thr200 and the SO_2_ group. Additionally, the nitrogen atom of the sulfonamide group formed a coordination bond with the zinc ion. The naphthyl moiety of 5FL4-ref formed interactions with Val130 and Leu134, and the heterocycle moiety of 5FL4-ref interacted with Gln92 and Thr199 via hydrogen and hydrophobic bond, respectively. The residues with which compounds (**4**), (**7**) and (**11**) interacted were essentially identical. Compounds (**4**), (**7**) and (**11**) also formed interactions with the zinc cation by coordinating the nitrogen atom of the sulfonamide. However, the interaction with the important residue Val130 was formed by the fragment of thiazolidinone in compounds (**4**), (**7**) and (**11**) (part B, C and D of Figure 2). At the same time, carbonyl groups in the thiazolidinone fragments of compounds (**4**) and (**11**) also formed hydrogen bonds with Gln92. Due to the rigidity of the heterocycle fragments in molecules, the substituted phenyl groups of compounds (**4**) and (**11**) achieved the same torsion angle in the active site when bound to Gln92. Due to the presence of trifluoromethyl, the conformation of compound (**7**) was slightly different from that of other two compounds and did not form interaction with residues Leu91 and Gln92, which may be the reason for the weaker activity of compound (**7**) than that of compounds (**4**) and (**11**). In the enzyme inhibition tests, the series of compounds also showed quite good inhibitory activity against *h*CA II. To illustrate the reasons, compounds (**4**), (**7**) and (**11**) and 4IWZ-ref were chosen for docking studies into the active site of *h*CA II. The results were shown in Figure 3.

Although most of the compounds showed good inhibitory activity, some of them showed little activity. To explain this phenomenon, the dominant conformations of compounds (**2**), (**3**), (**12**) and (**13**) in CA IX were superimposed. As shown in Figure 4, compounds (**2**) and (**3**) with better activity maintained almost the same conformation in the active site, whereas compounds (**12**) and (**13**) underwent considerable torsion, which led to an increase in the distance between the nitrogen atom of the sulfonamide group and the zinc ion. The increased distance may be the reason for the poor activities of these two compounds.

## 3. Materials and Methods

### 3.1. Chemistry

All the reagents (Energy Chemical, Shanghai, China) were used without further purification unless otherwise specified. Solvents were dried and redistilled prior to use in the usual manner. Analytical TLC was per-formed using silica gel HF_254_ (Qingdao Haiyang Chemical, Qingdao, China). Preparative column chromatography was performed with silica gel H. Melting points were obtained on a Büchi melting point B-540 apparatus. ^1^H and ^13^C NMR spectra (details of raw data for compounds, see Appendix A) were recorded on a Bruker ARX 600 MHz or 400 MHz spectrometer (Bruker, Zurich, Switzerland) HRMS were obtained on an Agilent ESI-QTOF instrument (Agilent, Santa Clara, CA, USA).

#### 3.1.1. 4-((3-phenylthioureido)methyl)benzenesulfonamide (**1**)

Compound (**1**) was prepared according to the literature procedures [28]. To a solution of mafenide hydrochloride (1.0 g, 4.5 mmol) and isothiocyanatobenzene (0.54 mL, 4.5 mmol) in DMF (10 mL) sodium bicarbonate (0.42 g, 5.0 mmol) and molecular sieves (4 Å, 1.0 g) were added. The mixture was stirred under room temperature for 4 h. After the molecular sieves were filtered off, 15 mL of water was added. The mixture was stirred vigorously to form white solids, which were filtered off to afford compound (**1**) (0.99 g, 68%).

#### 3.1.2. (Z)-4-((4-oxo-2-(phenylimino)thiazolidin-3-yl)methyl)benzenesulfonamide (**2**)

To a mixture of compound (**1**) (0.55 g, 1.7 mmol) and chloroacetic acid (0.24 g, 2.6 mmol) in EtOH (20 mL), sodium acetate anhydrous (0.28 g, 3.4 mmol) was added. The mixture was stirred at 78 °C for 12 h. The solvent was removed under reduced pressure, and the resulting residue was dissolved in ethyl acetate and washed by water. Then, ethyl acetate was removed under reduced pressure, affording light yellow solid, which was recrystallized from ethanol to give compound (**2**) as white powder (0.28 g, 45%). m.p. 179.2–180.3 °C; ^1^H-NMR (DMSO-*d_6_*, 600 MHz) δ 7.81 (d, *J* = 8.1 Hz, 2H), 7.55 (d, *J* = 8.2 Hz, 2H), 7.42–7.29 (m, 4H), 7.13 (t, *J* = 7.4 Hz, 1H), 6.90 (d, *J* = 7.4 Hz, 2H), 4.99 (s, 2H), 4.13 (s, 2H); ^13^C NMR (DMSO-*d_6_*, 151 MHz) δ 172.49, 155.47, 148.28, 143.66, 140.48, 129.79, 128.49, 126.32, 124.88, 121.33, 45.45, 33.10; HRMS (ESI): Calcd. for [M + Na]^+^ C_16_H_15_N_3_O_3_S_2_: 361.0555, Found 361.0548.

#### 3.1.3. General Procedure of Synthesis of 4-(((Z)-5-((Z)-benzylidene)-4-oxo-2-(phenylimino)thiazolidin-3-yl)methyl)benzenesulfonamide hybrids (**3**–**14**)

A mixture of compound (**2**) (0.14 g, 0.39 mmol), different aldehydes (0.39 mmol), piperidine (0.43 mmol, 0.040 mL), and ethanol (25 mL) were refluxed for 5 h. The reaction mixture was cooled to room temperature. The solvent was removed under reduced pressure, and the resulting residue was purified by column chromatography (150:1, dichloromethane-methanol) to give compounds (**3–14**), respectively.

*4-(((Z)-5-((Z)-benzylidene)-4-oxo-2-(phenylimino)thiazolidin-3-yl)methyl)benzenesulfonamide (**3**)*: Light yellow powder, m.p. 227.0–229.3 °C, 42% yield; ^1^H-NMR (CD_3_CN, 600 MHz) δ 7.89 (d, *J* = 8.4 Hz, 2H), 7.79 (s, 1H), 7.66 (d, *J* = 8.4 Hz, 2H), 7.53 (d, *J* = 7.4 Hz, 2H), 7.48 (dd, *J* = 10.2, 4.8 Hz, 2H), 7.43 (t, *J* = 7.9 Hz, 3H), 7.24 (t, *J* = 7.5 Hz, 1H), 7.05–7.00 (m, 2H), 5.68 (s, 2H), 5.22 (s, 2H); ^13^C-NMR (CD_3_CN, 151 MHz) δ 166.87, 150.88, 148.55, 143.13, 141.45, 134.21, 131.29, 130.60, 130.42, 130.07, 129.74, 129.17, 126.86, 125.59, 122.15, 121.60, 46.16; HRMS (ESI): Calcd. for [M + Na]^+^ C_23_H_19_N_3_O_3_S_2_: 449.0868, Found 449.0872.

*4-(((Z)-5-((Z)-3-fluorobenzylidene)-4-oxo-2-(phenylimino)thiazolidin-3-yl)methyl)benzenesulfonamide (**4**)*: Light yellow powder, m.p. 226.9–227.6 °C, 45% yield; ^1^H-NMR (CD_3_CN, 600 MHz) δ 7.89 (d, *J* = 8.4 Hz, 2H), 7.72 (s, 1H), 7.70–7.64 (m, 3H), 7.59 (dd, *J* = 8.0, 0.9 Hz, 1H), 7.50 (d, *J* = 7.9 Hz, 1H), 7.44 (t, *J* = 7.9 Hz, 2H), 7.40 (t, *J* = 7.9 Hz, 1H), 7.25 (t, *J* = 7.4 Hz, 1H), 7.03 (d, *J* = 7.4 Hz, 2H), 5.68 (s, 2H), 5.22 (s, 2H); ^13^C-NMR (CD_3_CN, 151 MHz) δ 166.76, 150.52, 148.57, 143.33, 141.46, 136.65, 133.33, 131.68, 130.26, 129.62, 129.39, 128.72, 127.01, 125.85, 124.11, 123.26, 121.70, 46.43; HRMS (ESI): Calcd. for [M + Na]^+^ C_23_H_18_FN_3_O_3_S_2_: 467.0774, Found 467.0779.

*4-(((Z)-5-((Z)-3-chlorobenzylidene)-4-oxo-2-(phenylimino)thiazolidin-3-yl)methyl)benzenesulfonamide (**5**)*: Light yellow powder, m.p. 218.9–220.9 °C, 41% yield; ^1^H-NMR (CD_3_CN, 600 MHz) δ 7.89 (d, *J* = 8.4 Hz, 2H), 7.72 (s, 1H), 7.66 (d, *J* = 8.4 Hz, 2H), 7.52 (s, 1H), 7.47–7.41 (m, 5H), 7.25 (t, *J* = 7.5 Hz, 1H), 7.02 (dd, *J* = 8.3, 0.9 Hz, 2H), 5.68 (s, 2H), 5.22 (s, 2H); ^13^C-NMR (CD_3_CN, 151 MHz) δ 165.23, 149.00, 147.03, 141.78, 139.92, 134.86, 133.65, 129.94, 128.83, 128.81, 128.15, 127.84, 126.84, 125.47, 124.30, 122.59, 120.15, 44.89; HRMS (ESI): Calcd. for [M + Na]^+^ C_23_H_18_ClN_3_O_3_S_2_: 483.0478, Found 483.0492.

*4-(((Z)-5-((Z)-3-bromobenzylidene)-4-oxo-2-(phenylimino)thiazolidin-3-yl)methyl)benzenesulfonamide (**6**)*: Light yellow powder, m.p. 217.7–220.3 °C, 48% yield; ^1^H-NMR (CD_3_CN, 600 MHz) δ 7.89 (d, *J* = 8.4 Hz, 2H), 7.72 (s, 1H), 7.70–7.64 (m, 3H), 7.59 (dd, *J* = 8.0, 0.9 Hz, 1H), 7.50 (d, *J* = 7.9 Hz, 1H), 7.44 (t, *J* = 7.9 Hz, 2H), 7.40 (t, *J* = 7.9 Hz, 1H), 7.25 (t, *J* = 7.4 Hz, 1H), 7.03 (d, *J* = 7.4 Hz, 2H), 5.68 (s, 2H), 5.22 (s, 2H); ^13^C-NMR (CD_3_CN, 151 MHz) δ 166.76, 150.52, 148.57, 143.33, 141.46, 136.65, 133.33, 131.68, 130.26, 129.62, 129.39, 128.72, 127.01, 125.85, 124.11, 123.26, 121.70, 46.43; HRMS (ESI): Calcd. for [M + Na]^+^ C_23_H_18_BrN_3_O_3_S_2_: 526.9973, Found 526.9967.

*4-(((Z)-4-oxo-2-(phenylimino)-5-((Z)-3-(trifluoromethyl)benzylidene)thiazolidin-3-yl)methyl)benzenesulfonamide (**7**)*: Light yellow powder, m.p. 226.6-227.4 °C, 44% yield; ^1^H-NMR (CD_3_CN, 600 MHz) δ 7.89 (d, *J* = 8.4 Hz, 2H), 7.72 (s, 1H), 7.66 (d, *J* = 8.4 Hz, 2H), 7.52 (s, 1H), 7.47–7.41 (m, 5H), 7.25 (t, *J* = 7.5 Hz, 1H), 7.02 (dd, *J* = 8.3, 0.9 Hz, 2H), 5.68 (s, 2H), 5.22 (s, 2H); ^13^C-NMR (CD_3_CN, 151 MHz) δ 165.23, 149.00, 147.03, 141.78, 139.92, 134.86, 133.65, 129.94, 128.92–128.62, 128.15, 127.84, 126.84, 125.47, 124.30, 122.59, 120.15, 44.89; HRMS (ESI): Calcd. for [M + Na]^+^ C_24_H_18_F_3_N_3_O_3_S_2_: 517.0742, Found 517.0760.

*4-(((Z)-5-((Z)-3-hydroxybenzylidene)-4-oxo-2-(phenylimino)thiazolidin-3-yl)methyl)benzenesulfonamide (**8**)*: Yellow powder, m.p. 268.2–269.3 °C, 30.30% yield; ^1^H-NMR (DMSO-*d_6_*, 400 MHz) δ 9.83 (s, 1H), 7.83 (d, *J* = 8.3 Hz, 2H), 7.70 (s, 1H), 7.59 (d, *J* = 8.3 Hz, 2H), 7.48 – 7.24 (m, 5H), 7.20 (t, *J* = 7.4 Hz, 1H), 7.00 (t, *J* = 7.8 Hz, 3H), 6.92 (t, *J* = 1.8 Hz, 1H), 6.84 (dd, *J* = 8.1, 1.7 Hz, 1H), 5.17 (s, 2H); ^13^C-NMR (DMSO-*d_6_*, 101 MHz) δ 166.24, 158.35, 150.14, 147.99, 143.77, 140.29, 134.81, 131.58, 130.82, 130.02, 128.46, 126.46, 125.47, 121.84, 121.46, 121.11, 118.05, 115.96, 45.87; HRMS (ESI): Calcd. for [M + Na]^+^ C_23_H_19_N_3_O_4_S_2_: 465.0817, Found 465.0814.

*4-(((Z)-5-((Z)-4-fluorobenzylidene)-4-oxo-2-(phenylimino)thiazolidin-3-yl)methyl)benzenesulfonamide (**9**)*: Light yellow powder, m.p. 209.7–211.0 °C, 33% yield; ^1^H-NMR (CD_3_CN, 600 MHz) δ 7.92–7.83 (m, 2H), 7.77 (s, 1H), 7.66 (d, *J* = 8.5 Hz, 2H), 7.60 – 7.52 (m, 2H), 7.47–7.39 (m, 2H), 7.27–7.18 (m, 3H), 7.07–6.98 (m, 2H), 5.69 (s, 2H), 5.22 (s, 2H); ^13^C-NMR (CD_3_CN, 151 MHz) δ 166.82, 164.57, 162.91, 150.70, 148.54, 143.15, 141.42, 132.73, 130.75, 130.09, 129.20, 126.87, 125.62, 121.92, 121.60, 116.87, 116.72, 46.20; HRMS (ESI): Calcd. for [M + Na]^+^ C_23_H_18_FN_3_O_3_S_2_: 467.0774, Found 467.0771.

*4-(((Z)-5-((Z)-4-hydroxybenzylidene)-4-oxo-2-(phenylimino)thiazolidin-3-yl)methyl)benzenesulfonamide (**10**)*: Yellow powder, m.p. 176.3–178.2 °C, 30% yield; ^1^H-NMR (DMSO-*d_6_*, 400 MHz) δ 7.83 (d, *J* = 8.3 Hz, 2H), 7.68 (s, 1H), 7.57 (d, *J* = 8.3 Hz, 2H), 7.40 (t, *J* = 7.8 Hz, 2H), 7.35 (d, *J* = 8.7 Hz, 2H), 7.19 (t, *J* = 7.4 Hz, 1H), 7.00 (d, *J* = 7.4 Hz, 2H), 6.84 (d, *J* = 8.7 Hz, 2H), 5.15 (s, 2H); ^13^C-NMR (DMSO-*d_6_*, 101 MHz) δ 166.52, 162.30, 150.49, 148.20, 143.72, 140.50, 132.79, 132.24, 129.98, 128.37, 126.45, 125.27, 123.24, 121.51, 117.31, 115.24, 46.47; HRMS (ESI): Calcd. for [M + Na]^+^ C_23_H_19_N_3_O_4_S_2_: 465.0817, Found 465.0824.

*4-(((Z)-5-((Z)-4-methoxybenzylidene)-4-oxo-2-(phenylimino)thiazolidin-3-yl)methyl)benzenesulfonamide (11)*: Light yellow powder, m.p. 210.6–211.8 °C, 39% yield; ^1^H-NMR (CD_3_CN, 600 MHz) δ 7.88 (d, *J* = 8.4 Hz, 2H), 7.74 (s, 1H), 7.66 (d, *J* = 8.4 Hz, 2H), 7.43 (t, *J* = 7.9 Hz, 2H), 7.23 (t, *J* = 7.4 Hz, 1H), 7.14–7.08 (m, 2H), 7.03 (dd, *J* = 7.8, 2.9 Hz, 3H), 5.68 (s, 1H), 5.22 (s, 2H), 3.85 (s, 3H), 3.81 (s, 3H); ^13^C-NMR (CD_3_CN, 151 MHz) δ 167.04, 161.70, 151.08, 148.70, 143.09, 141.57, 132.43, 131.25, 130.05, 129.13, 126.85, 126.70, 125.50, 121.63, 119.00, 115.24, 55.79, 46.08; HRMS (ESI): Calcd. for [M + Na]^+^ C_24_H_21_N_3_O_4_S_2_: 479.0973, Found 479.0981.

*4-(((Z)-5-((Z)-4-(benzyloxy)benzylidene)-4-oxo-2-(phenylimino)thiazolidin-3-yl)methyl)benzenesulfonamide (12)*: Light yellow powder, m.p. 219.6–221.3 °C, 48% yield; ^1^H-NMR (CD_3_CN, 600 MHz) δ 7.88 (d, *J* = 8.4 Hz, 2H), 7.74 (s, 1H), 7.66 (d, *J* = 8.4 Hz, 2H), 7.43 (t, *J* = 7.9 Hz, 2H), 7.23 (t, *J* = 7.4 Hz, 1H), 7.14–7.08 (m, 2H), 7.03 (dd, *J* = 7.8, 2.9 Hz, 3H), 5.68 (s, 1H), 5.22 (s, 2H), 3.85 (s, 3H), 3.81 (s, 3H); ^13^C-NMR (CD_3_CN, 151 MHz) δ 167.00, 151.56, 151.00, 149.88, 148.60, 143.09, 141.59, 131.54, 130.04, 129.13, 126.91, 125.51, 123.57, 121.65, 119.34, 114.33, 112.38, 57.51, 56.01, 46.09; HRMS (ESI): Calcd. for [M + Na]^+^ C_30_H_25_N_3_O_4_S_2_: 555.1286, Found 555.1279.

*4-(((Z)-5-((Z)-4-isopropylbenzylidene)-4-oxo-2-(phenylimino)thiazolidin-3-yl)methyl)benzenesulfonamide (**13**)*: Light yellow powder, m.p. 200.7–201.6 °C, 49% yield; ^1^H NMR (CD_3_CN, 600 MHz) δ 7.92–7.84 (m, 2H), 7.76 (s, 1H), 7.65 (d, *J* = 8.5 Hz, 2H), 7.48–7.40 (m, 4H), 7.36 (d, *J* = 8.3 Hz, 2H), 7.26–7.20 (m, 1H), 7.06–6.98 (m, 2H), 5.68 (s, 2H), 5.21 (s, 2H), 2.95 (hept, *J* = 6.9 Hz, 1H), 1.23 (d, *J* = 6.9 Hz, 6H); ^13^C-NMR (CD_3_CN, 151 MHz) δ 166.94, 152.13, 150.99, 148.60, 143.11, 141.49, 131.83, 131.38, 130.63, 130.05, 129.17, 127.83, 126.85, 125.54, 121.61, 121.01, 46.11, 34.38, 23.46; HRMS (ESI): Calcd. for [M + Na]^+^ C_26_H_25_N_3_O_3_S_2_: 491.1337, Found 493.1343.

*4-(((Z)-5-((Z)-3,4-dimethoxybenzylidene)-4-oxo-2-(phenylimino)thiazolidin-3-yl)methyl)benzenesulfonamide (**14**)*: Light yellow powder, m.p. 249.8–250.9 °C, 41% yield; ^1^H-NMR (CD_3_CN, 600 MHz) δ 7.88 (d, *J* = 8.4 Hz, 2H), 7.74 (s, 1H), 7.66 (d, *J* = 8.4 Hz, 2H), 7.43 (t, *J* = 7.9 Hz, 2H), 7.23 (t, *J* = 7.4 Hz, 1H), 7.14–7.08 (m, 2H), 7.03 (dd, *J* = 7.8, 2.9 Hz, 3H), 5.68 (s, 1H), 5.22 (s, 2H), 3.85 (s, 3H), 3.81 (s, 3H); ^13^C-NMR (CD_3_CN, 151 MHz) δ 167.00, 151.56, 151.00, 149.88, 148.60, 143.09, 141.59, 131.54, 130.04, 129.13, 126.91, 125.51, 123.57, 121.65, 119.34, 114.33, 112.38, 57.51, 56.01, 46.09; HRMS (ESI): Calcd. for [M + Na]^+^ C_25_H_23_N_3_O_5_S_2_: 509.1079, Found 509.1092.

### 3.2. hCA Enzyme Inhibition Assays

Carbonic anhydrases are able to catalyze the conversion 4-nitrophenyl acetate (4-NPA) to 4-nitrophenol. According to the method described previously by Verpoorte, the rate of this reaction is monitored spectrophotometrically, at 405 nm, with a Perkin Elmer Envision 2104 plate reader [30,31] (Perkin Elmer, Waltham, MA, USA). 1X assay buffer consisted of 15 mmol/L 4-(2-hydroxyethyl)-1-piperazineethanesulfonic acid (HEPES) (pH = 7.40) 0.01% tetraethylene glycol monododecyl ether (Brij) and 100 mmol/L NaCl. Recombinant human carbonic anhydrase II and IX was commercially available (Sino Biological Inc, Beijing, China) and prepared in 1X assay buffer with a concentration of 11.10 ng/μL. Then, 18 μL of enzyme solution was transferred into 384-well assay plates in triplicate. Stock solutions of the inhibitor (10.00 mmol/L) were prepared using DMSO as solvent, and then diluted 1:3 with DMSO. Ten different inhibitor concentrations were used: 600.00 μmol/L, 200.00 μmol/L, 66.67 μmol/L, 22.22 μmol/L, 7.41 μmol/L, 2.47 μmol/L, 0.82 μmol/L, 0.27 μmol/L, 0.091 μmol/L, 0.03 μmol/L, and 0 μmol/L. A volume of 2.00 μL of each inhibitor was added into the assay solution. All compounds were allowed to incubate with the enzyme for 15 min at 25 °C to form the Enzyme-Inhibitor (E-I) complex. After that, substrate 4-NPA (1.00 mmol/L, 20.00 μL, Sigma-Aldrich, St. Louis, MI, USA) was added into the E-I complex solution and incubated for 90 min at 25 °C. The absorbance of each compound was measured with Envision 2104 plate reader. The inhibitor AZM and SLC-0111 were used as standards to investigate the inhibitory activity of these compounds.

### 3.3. Single-Crystal Structure

The configuration of the target compound was determined by the single-crystal X-ray diffraction (SuperNova, Rigaku, Tokyo, Japan) of compound (**13**). The compound crystallized in a centrosymmetric space groups as a triclinic system. The resulting structure is shown in Figure 5 (CCDC Number: 1888237). The experiment showed that the double bonds between N2-C10 and C9-C17 are both in the Z configuration.

### 3.4. Preparation of Compound **2**–**14** for Docking Studies

Molecular docking calculations using the ligand molecules with CA IX (PDB code: 5FL4) [32] and CA II (PDB: 4IWZ) [33] were conducted using the modified version of Autodock 4(Zn) [34,35]. A grid box with a grid spacing of 0.375 Å was generated to define the binding pocket. AutoGrid 4 was used as the auxiliary program to generate affinity grid fields with the modified forcefield to generate maps files. Compound structures were built and minimized with the Accelrys Discovery Studio 3.0 software package [36] with flexible torsions assigned, and all dihedral angles were able to freely rotate. We applied the Lamarckian genetic algorithm to determine the appropriate binding positions, orientations, and conformations of the ligands. The optimized parameters were as follows: The maximum number of energy evaluations was increased to 25,000,000 per run, the iterations of Solis and Wets local search were 3000, the number of individuals in the population was 300, and the number of generations was 25. The results differing by <2 Å in a positional root mean square deviation were clustered together. In each group, the lowest binding energy configuration with the highest percentage frequency was selected as the group representative. All other parameters were maintained as default parameters.

## 4. Conclusions

A series of benzenesulfonamide-thiazolidinone derivatives was synthesized and evaluated for the ability to inhibit *h*CA II and IX in vitro. Although the target compounds showed less selectivity between CA II and IX, the inhibitory activities of most derivatives were equivalent to AZM but were much higher than that of the precursor compound. The conclusion of SARs indicates that the introduction of heterocyclic rings in the molecules can increase CA inhibitory activity and is worthy of being further explored. In addition, the molecular docking results of *h*CA II and IX indicated that the benzenesulfonamide derivatives containing thiazolidinone were structurally rigid and extended into the active sites of CAs with relatively fixed conformations. Such a rigid and relatively fixed conformation could interact with the key residue Val130 in the active site of CA IX, and without interacting with Phe130 in CA II. Further exploration around this point of view is expected to help improve the CA IX/II selectivity of this class of structure.

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
