# Peer review of "Synthesis, Molecular Docking Analysis, and Carbonic Anhydrase Inhibitory Evaluations of Benzenesulfonamide Derivatives Containing Thiazolidinone"

_molecules, 2019, doi:10.3390/molecules24132418_

Round 1
Reviewer 1 Report
The manuscript “Synthesis, molecular docking analysis, and carbonic anhydrase inhibitory evaluations of benzenesulfonamide derivatives containing thiazolidinone” by Zuo-peng Zhang et al reports the investigation on the role of the thiazolidinone moiety in new thirteen compounds designed as inhibitors of the isoforms of CA IX and CA II. The authors' clear intent was to extend the applicability of thiazolidinones to CA inhibitors, in particular considering the versatility of this heterocyclic ring in medicinal chemistry. The synthesized compounds have an inhibitory activity comparable to reference compound SLC-0111, currently present in clinical phase and much lower than AZM; this is explained by means of docking study using the dominant conformations of investigated compounds. In the opinion of this referee the manuscript fits with criteria required by Molecules however it is necessary to improve and update the bibliographic references, moreover it is necessary to indicate them according to the notes for the authors. (reference 25 does not comply with the magazine's requests)
Some suggested references:
JOURNAL OF ENZYME INHIBITION AND MEDICINAL CHEMISTRY 2019, VOL. 34, NO. 1, 117–123 https://doi.org/10.1080/14756366.2018.1532419
Metabolites 2018, 8, 19; doi:10.3390/metabo8010019 www.mdpi.com/journal/metabolites
For clarity it is advisable to indicate the compound U104 (line78 page 3) with the abbreviation SLC-0111 in which it appears in the table 1.
After minor revision the manuscript could be published ; this referee would like to review the corrected and implemented manuscript.
Author Response
Many thanks for the reviewers’ comments concerning our manuscript entitled “Synthesis, molecular docking analysis, and carbonic anhydrase inhibitory evaluations of benzenesulfonamide derivatives containing thiazolidinone” (ID: molecules-528562). Those comments are all valuable and very helpful for revising and improving our paper, as well as the important guiding significance to our researches. We have studied comments carefully and made essential corrections which we hope meet with approval. The main corrections in the paper and the responds to the reviewer’s comments are as following.
Part A (to Reviewer 1)
⑴ In the opinion of this referee the manuscript fits with criteria required by Molecules however it is necessary to improve and update the bibliographic references, moreover it is necessary to indicate them according to the notes for the authors. (reference 25 does not comply with the magazine's requests)
Many thanks for the reviewer’s suggestion, we have added more relevant literatures on recent CA study in the Introduction and References parts.
⑵ For clarity it is advisable to indicate the compound U104 (line78 page 3) with the abbreviation SLC-0111 in which it appears in the table 1.
Many thanks for the reviewer’s suggestion. All the name of U-104 in our paper have been replaced with SLC-0111.

Reviewer 2 Report
The paper has described the synthesis of benzenesulfonamide derivatives with thiazolidinone ring and their biological evaluation with carbonic anhydrase. The synthetic strategy is very simple and several benzenesulfonamide derivatives have been obtained in short synthetic steps. In the inhibitory assay of their derivatives with carbonic anhydrase, the authors showed that synthetic compounds have inhibitory potency for carbonic hydrase IX, which is a promising target for cancer diagnosis and treatment. In addition, in molecular modeling, the authors predicted binding mode of benzenesulfonamide derivatives by comparison with more active and less active derivarives. The results are potentially useful not only for pharmaceutical scientist but also biological scientist, and the paper is valuable for publication in Molecules. However, the following points should be considered for the revision of this paper to be accepted.
(1) The authors should describe the rationale why the derivatives of thiazolidinone at 2-, 3-, and 5-position were examined. Various aryl isothiocyanates are commercially available and the derivatives based on compound 1 and 2 are also considered as good candidates for inhibiting CA.
(2) The authors should describe the reason why CA II was used for comparison with CA IX.
(3) Regarding compounds, yields are not so high. Are there any regio- and stereoisomers in the reaction? If there are, the authors should describe how their structures were determined. For example, the regioisomer (cyclized with N atom of aniline moiety) is considered for compound 2, and EZ stereoisomers are considered for compound 3-14. Although X-ray structural analysis for compound 13 has been performed, the authors should describe the reason for other compounds, too.
(4) In molecular modeling, the authors should discuss the relationship between IC50 and delta G.
Minor points,
Line 38-39: "position C1," "position C3," "positions C2, C4 or C5" >> should be "1-position," "3-position," "2-, 4-, or 5-positions," respectively.
Line 78: "AZM and U104" >> Compound name and explanations are needed.
Figure 2: hard to see name of amino acids >> should be enlarged.
Figure 3: hard to see name of amino acids and hard to see compounds >> should be enlarged and color of protein backborn should be change to lightgray.
Line 142: "per-formed" >> "performed"
Line 143: "chroma-tography" >> "chromatography"
Material and Methods: should be check significant digit (compound amounts, molar, and yield).
Line 171, 178, 186, 194, 202, 210, 218, 225, 233, 241, 249, and 257: "4.-(((Z)" >> "4-(((Z)" (remove period)
Author Response
Many thanks for the reviewers’ comments concerning our manuscript entitled “Synthesis, molecular docking analysis, and carbonic anhydrase inhibitory evaluations of benzenesulfonamide derivatives containing thiazolidinone” (ID: molecules-528562). Those comments are all valuable and very helpful for revising and improving our paper, as well as the important guiding significance to our researches. We have studied comments carefully and made essential corrections which we hope meet with approval. The main corrections in the paper and the responds to the reviewer’s comments are as following.
Part B (to Reviewer 2)
⑴ The authors should describe the rationale why the derivatives of thiazolidinone at 2-, 3-, and 5-position were examined. Various aryl isothiocyanates are commercially available and the derivatives based on compound 1 and 2 are also considered as good candidates for inhibiting CA.
The 2, 3, 5 -thiazolidinone is a type of structure obtained by the synthetic method used herein, and it also exhibits certain activities as a lead compound in the in vitro activity test. Therefore, a series of benzenesulfonamide derivatives containing thiazolidinone were synthesized with 2,3,5-thiazolidinone as a substructure, under the direction of molecular hybrid strategy. Many thanks for the reviewer’s valuable suggestion, we will examine the application of other aryl isothiocyanates in carbonic anhydrase inhibitors.
⑵ The authors should describe the reason why CA II was used for comparison with CA IX.
Many thanks for the reviewer’s suggestion. As a subtype of the CA family, CA II is widely distributed in the body and participates in various physiological activities. An ideal CA IX inhibitor may not affect CA II or a little. However, our compounds lack this selectivity, which is one of drawbacks of this study. Anyway, our work offers a new class of structure that can increase CA inhibitory activity and is worthy of being further explored. The reason why CA II is compared with CA IX has been supplemented in the Introduction part of the revised manuscript.
⑶ Regarding compounds, yields are not so high. Are there any regio- and stereoisomers in the reaction? If there are, the authors should describe how their structures were determined. For example, the regioisomer (cyclized with N atom of aniline moiety) is considered for compound 2, and EZ stereoisomers are considered for compound 3-14. Although X-ray structural analysis for compound 13 has been performed, the authors should describe the reason for other compounds, too.
Many thanks for the reviewer’s questions. In the preparation of compound 3-14, there was only one main product was detected by TLC. When TLC monitoring was repeated with different developing solvent systems, no isomer was found. The reason for the relative lower yield was the remainder of parts of raw materials and the loss of the product due to the adsorption by silica gel during the column chromatography operation.
⑷ In molecular modeling, the authors should discuss the relationship between IC50 and delta G.
Many thanks for the reviewer’s suggestion. We have supplemented docking study for all the compounds 2-14, and added the binding score results in Table 1.
⑸ Line 38-39: "position C1," "position C3," "positions C2, C4 or C5" >> should be "1-position," "3-position," "2-, 4-, or 5-positions," respectively. Line 78: "AZM and U104" >> Compound name and explanations are needed. Figure 2: hard to see name of amino acids >> should be enlarged. Figure 3: hard to see name of amino acids and hard to see compounds >> should be enlarged and color of protein backborn should be change to lightgray. Line 142: "per-formed" >> "performed"
Line 143: "chroma-tography" >> "chromatography" Material and Methods: should be check significant digit (compound amounts, molar, and yield). Line 171, 178, 186, 194, 202, 210, 218, 225, 233, 241, 249, and 257: "4.-(((Z)" >> "4-(((Z)" (remove period)
Thanks for the reviewer’s careful examination. According to your suggestion, the above errors have been corrected in the revised manuscript.
Reviewer 3 Report
The manuscript by Zuo-peng Zhang et al. “Synthesis, molecular docking analysis, and carbonic anhydrase inhibitory evaluations of benzenesulfonamide derivatives containing thiazolidinone” reports the synthesis and hCA inhibitory activity of 12 thiazolidinone derivatives. The enzyme inhibition was assayed by monitoring the conversion of 4-nitrophenyl acetate to 4-nitrophenol. New compounds were characterized by conventional techniques (NMR, HRMS, melting point determination). In addition, molecular structure of one of the synthesized derivatives was determined by single crystal X-ray diffraction. The manuscript is well written and may be of interest to medicinal chemistry community.
I have a few remarks:
1. The authors present the results of molecular docking studies for only three out of twelve synthesized compounds. I think molecular docking results would have much more value if all 12 molecules were studied and some quantitative scoring functions would be given for each of them and compared to experimental inhibitory activity.
2. No analytical data is presented for compound 1. If this compound was prepared using the literature procedure (it is mentioned in several papers on CA inhibitors), citation should be given. If the compound was not fully characterized before, I suggest to give full spectral data in the experimental part.
3. Yields should be rounded up to integers (e.g. 40 %, not 40.45 %).
Author Response
Many thanks for the reviewers’ comments concerning our manuscript entitled “Synthesis, molecular docking analysis, and carbonic anhydrase inhibitory evaluations of benzenesulfonamide derivatives containing thiazolidinone” (ID: molecules-528562). Those comments are all valuable and very helpful for revising and improving our paper, as well as the important guiding significance to our researches. We have studied comments carefully and made essential corrections which we hope meet with approval. The main corrections in the paper and the responds to the reviewer’s comments are as following.
Part C (to Reviewer 3)
⑴ The authors present the results of molecular docking studies for only three out of twelve synthesized compounds. I think molecular docking results would have much more value if all 12 molecules were studied and some quantitative scoring functions would be given for each of them and compared to experimental inhibitory activity.
Many thanks for the reviewer’s suggestion. We performed the molecular docking studies after the bioactivity evaluation to illustrate the possible reason for the difference among the activity. According to the reviewer’s suggestion, we have supplemented docking study for all the compounds 2-14, and added the binding score results in Table 1. It should be emphasized that there was a certain relationship between the binding scores with the value of IC50. The compound with better activity owned more negative score value. This conclusion also supports our previously summarized SARs.
⑵ No analytical data is presented for compound 1. If this compound was prepared using the literature procedure (it is mentioned in several papers on CA inhibitors), citation should be given. If the compound was not fully characterized before, I suggest to give full spectral data in the experimental part.
Many thanks for the reviewer’s suggestion. The synthetic method for compound 1 has been reported in the literature, and the citation has been supplemented in the revised manuscript.
⑶ Yields should be rounded up to integers (e.g. 40 %, not 40.45 %).
Thanks for the reviewer’s careful examination. All the data for yields has been rounded up to integers in the revised manuscript.
Round 2
Reviewer 2 Report
The revised paper has been improved. This paper is recommended to be published.
Reviewer 3 Report
The authors took great care to improve the manuscript taking into account the reviewers comments and I can now recommend the manuscript for publication